# The Effects of the Red River Jig on the Wholistic Health of Adults in Saskatchewan

**DOI:** 10.3390/ijerph22081225

**Published:** 2025-08-06

**Authors:** Nisha K. Mainra, Samantha J. Moore, Jamie LaFleur, Alison R. Oates, Gavin Selinger, Tayha Theresia Rolfes, Hanna Sullivan, Muqtasida Fatima, Heather J. A. Foulds

**Affiliations:** 1College of Kinesiology, University of Saskatchewan, Saskatoon, SK S7N 5B2, Canadaalison.oates@usask.ca (A.R.O.); tayha.rolfes@usask.ca (T.T.R.); has003@mail.usask.ca (H.S.); muf506@mail.usask.ca (M.F.); 2College of Medicine, University of Saskatchewan, Saskatoon, SK S7N 5A5, Canada

**Keywords:** Indigenous Peoples, wholistic medicine, dancing, cultural anthropology, exercise, physical fitness

## Abstract

The Red River Jig is a traditional Métis dance practiced among Indigenous and non-Indigenous Peoples. While exercise improves physical health and fitness, the impacts of cultural dances on wholistic health are less clear. This study aimed to investigate the psychosocial (cultural and mental), social, physical function, and physical fitness benefits of a Red River Jig intervention. In partnership with Li Toneur Nimiyitoohk Métis Dance Group, Indigenous and non-Indigenous adults (*N* = 40, 39 ± 15 years, 32 females) completed an 8-week Red River Jig intervention. Social support, cultural identity, memory, and mental wellbeing questionnaires, seated blood pressure and heart rate, weight, pulse-wave velocity, heart rate variability, baroreceptor sensitivity, jump height, sit-and-reach flexibility, one-leg and tandem balance, and six-minute walk test were assessed pre- and post-intervention. Community, family, and friend support scores, six-minute walk distance (553.0 ± 88.7 m vs. 602.2 ± 138.6 m, *p =* 0.002), jump, leg power, and systolic blood pressure low-to-high-frequency ratio increased after the intervention. Ethnic identity remained the same while affirmation and belonging declined, leading to declines in overall cultural identity, as learning about Métis culture through the Red River Jig may highlight gaps in cultural knowledge. Seated systolic blood pressure (116.5 ± 7.3 mmHg vs. 112.5 ± 10.7 mmHg, *p* = 0.01) and lower peripheral pulse-wave velocity (10.0 ± 2.0 m·s^−1^ vs. 9.4 ± 1.9 m·s^−1^, *p* = 0.04) decreased after the intervention. Red River Jig dance training can improve social support, physical function, and physical fitness for Indigenous and non-Indigenous adults.

## 1. Introduction

Indigenous Peoples is a collective term for the original peoples of North America and their descendants, including First Nations, Inuit, and Métis People in Canada [1]. A legally recognized Indigenous Peoples in Canada, the Métis Nation is a specific nation of People with their own history, language, and culture [2]. Ongoing acts of colonization in Canada continue to affect the health and wellbeing of Indigenous populations, including elevated rates of chronic diseases such as cardiovascular disease (CVD), obesity, and diabetes [3]. From a Métis perspective, health and wellbeing are related to the geophysical environment, community, and culture [4]. Olvera (2008) described four main dimensions of health within Métis communities: cultural connectedness, mental health, social support, and physical health and fitness [5]. Within Métis contexts, health and wellbeing are understood to be multidimensional, encompassing psychosocial wellness, spirituality, community, culture, and mental health [6].

From a Métis perspective, health includes multiple aspects, including physical function, physical fitness, social health, and psychosocial wellbeing [1,7]. Psychosocial wellbeing, from a Métis worldview, includes mental wellbeing as well as cultural and spiritual connectedness [7]. Spiritual aspects refer to belief in God or a higher power with whom one may seek guidance [1]. Emotional aspects include feelings of anger, sadness, and joy, as well as physical symptoms of nervousness, stress, and anxiety [1]. Cultural aspects include feelings of identity and connectedness with one’s culture [5]. Mental aspects include feelings of self-efficacy, one’s mental wellbeing, and feelings of general wellbeing [5]. Physical aspects involve good nutrition, physical activity, and having the energy to undertake basic daily activities and bodily functioning [1].

Regular physical activity has many health benefits, such as weight and blood pressure reductions, and feelings of mental relief and positivity [8,9]. Leisure activities, such as sports and dancing, provide an escape from spaces where Métis People may be seen as outside the dominant culture and a unique opportunity to include cultural practices that benefit mental and physical health [5,10]. The Red River Jig is a traditional Métis fiddle tune and dance of the same name relevant to both Métis and First Nations communities [10]. The Red River Jig dance features two parts, alternating as the music moves to upper and lower pitches [10]. The double jig step (upper pitch) is alternated with ‘fancy steps’, which can be any other type of dance step, typically not repeated as a fancy step within the same dance (lower pitch) [10,11]. Dances commonly include at least four to eight changes (1.5–3 min), including at least four rounds each of jigging and fancy steps, though competitions can feature up to twenty-four changes or more (8 or more minutes) [10]. Originating during the time of the fur trade, Métis jigging brought together Métis, European, and First Nations Peoples to participate in gatherings, strengthening community ties and trade networks [10]. Métis culture and spirituality reflect importance of life cycles, community, environment, and religion [12]. Community dances were an important way of renewing and maintaining kinship and community [13,14]. The Red River Jig provides spatiotemporal conceptualization linking to Métis history and symbolizing survival, historical struggle, and continuation of Métis culture, which typically engages audience and community [10,13]. Métis jigging combines skill with spiritual and healing experiences for Indigenous Peoples [15]. Métis communities have long recognized the wholistic benefits of jigging related to the physical activity along with connection to culture and community; however, to date these wholistic health benefits of Red River Jigging have not be evaluated.

The primary objective of this study was to investigate the effects of an 8-week Red River Jigging intervention on cultural, mental, social and physical determinants of health. It is hypothesized that an 8-week Métis Red River Jigging intervention will effectively improve cultural, mental, social, and physical determinants of health.

## 2. Materials and Methods

### 2.1. Context, Community Partner, and Ethics

In present day Canada, Saskatchewan is one of ten provinces, located in the prairie region of the country [16]. These lands have been home to many First Nations Peoples, stretching back millennia to time immemorial [17]. Lands in Saskatchewan, extending to neighboring provinces, territories, and US states, have also been recognized as the Homeland of the Métis, since the birth of the Métis Nation more than 250 years ago [18]. Today, Métis People comprise 5.1% of the population in Saskatchewan, which includes 17.0% Indigenous Peoples [16]. The study adopted the Métis infinity symbol worldview to determine the effectiveness of the Red River Jig as a physical activity intervention [2]. This framework grounds the project in Métis culture, centering the infinity symbol of the Métis flag. In this project, the infinity symbol represents the bringing together of Indigenous and Western ways of knowing, being, and doing [2]. Similar to Mi’kmaw Two-Eyed Seeing and Cree Ethical Space approaches, this infinity symbol framework weaves together two knowledge systems to form a relational space for this work to occur [2,19,20]. Consistent with TCPS 2 Chapter 9, Métis data governance principles, and CARE (Collective benefit, Authority to control, Responsibility, and Ethics) principles, this study was conducted under the direction of Li Toneur Niimiyitoohk Métis dance group [21,22,23]. Community Advisors from Li Toneur Niimiyitoohk Métis dance group directed and approved all aspects of the research process, including study design, duration, instructors hired, wording and ordering of questionnaires, recruitment strategies, and data interpretation. Data are stored and maintained in the University of Saskatchewan servers under a Data Stewardship model. A survey of potential measures to evaluate in this study was sent to members of Li Toneur Niimiyitoohk Métis dance group, with measures achieving more than 50% support from survey responses included in the study measures and methods, as well as additional measures identified by the community through this survey (e.g., memory). Annual Métis Kitchen Dances, a traditional Métis cultural practice, were held to share preliminary study findings, gather input from Li Toneur Niimiyitoohk Métis dance group members, maintain relationships, and ensure benefits for Li Toneur Niimiyitoohk Métis dance group. Participants were recruited through Li Toneur Niimiyitoohk Métis dance group, Métis contacts and gatherings on the University of Saskatchewan campus, and Indigenous and non-Indigenous communities and contacts within the Saskatoon region. This research adhered to Human Ethics guidelines and was approved by the University of Saskatchewan Biomedical Research Ethics Board (Bio-3329) on 5 May 2022. Informed consent was attained from each person before voluntary participation in the study.

### 2.2. Intervention and Participants

Using a pre-experimental design, this study included a series of eight-week Red River Jigging dance classes held at the University of Saskatchewan campus or online in summer 2022 and the fall and winter terms of the 2022/23 through 2024/25 academic years. This series of dance interventions was primarily limited to fall/winter terms due to limited participant engagement in spring/summer. Subsequent iterations of the dance intervention engaged additional participants and provided opportunities for previous participants and members of Li Toneur Niimiyitoohk Métis dance group to attend dance classes. A total of eight interventions were held (four online) with new participants recruited each intervention to achieve the a priori sample size target of 40 participants post-tested to enable sub-sample analysis of blood pressures [24]. Participants completed a Canadian Society for Exercise Physiology Get Active Questionnaire before participation [25]. Participants were eligible to participate if they were at least 18 years old, did not have a history of heart disease (heart attack, stroke, bypass surgery, angina, etc.) or diabetes, and were cleared for unrestricted physical activity. Participants did not need to be a member of Li Toneur Niimiyitoohk Métis Dance Group or the University of Saskatchewan to participate in this study. Health determinants were measured individually within two weeks before and after the Red River Jigging classes at the Ron and Jane Graham Sport and Health Science Centre in Merlis Belsher Place at the University of Saskatchewan. Participants were provided with an honorarium of CAD $50 for each of the two health assessments. Each identical dance intervention comprised weekly one-hour sessions conducted either via Zoom or in person on the University of Saskatchewan campus or within the Saskatoon community. An experienced instructor from the Li Toneur Nimiyitoohk Métis Dance Group led the weekly dance classes. Participants were instructed to practice independently and asked to practice dancing at home an additional two times per week supported by YouTube videos and videos sent out by the instructors.

### 2.3. Study Procedures

Demographic measures, including age, sex, gender identity, ethnicity, educational status, and Red River Jigging history, were collected through a questionnaire. Age was reported in years, and sex was selected from options of male or female. Participants indicated their gender from a list of possible gender identities, including Woman, Man, Trans Man, Trans Woman, Two-Spirit, Genderqueer/Gender non-conforming, and self-specified responses. Ethnicity and educational status were evaluated using multiple choice questions from the Canadian Community Health Survey [26]. Red River Jigging history was evaluated through open-ended questions asking if participants had ever participated in Red River Jigging, currently danced the Red River Jig, danced the Red River Jig as part of a group, or participated in Red River Jigging competitions.

Cultural health was evaluated through the Multigroup Ethnic Identity Measure questionnaire, identifying scores for ethnic identity, affirmation and belonging, exploration, commitment, and overall cultural identity [27,28]. This 12-question tool uses a four-point Likert scale and averages scores of applicable questions, with higher scores indicating stronger cultural identity [27,28]. Mental health evaluations included general wellbeing through the WHO-5 wellbeing questionnaire, mental wellbeing through the Kessler K10 questionnaire, and self-efficacy through the General Self-Efficacy questionnaire [29,30,31]. The 5-question WHO-5 tool evaluates frequency of general feelings from at no time (0) to all of the time (5), with scores summed and ranging from 0 to 25, and higher scores indicating greater wellbeing [31]. The Kessler K10 asks about frequency of specific feelings over the past 30 days from none of the time (1) to all of the time (5) with scores summed and ranging from 10 to 50 [29]. Kessler K10 scores under 20 indicate mentally well individuals, with higher scores indicating greater mental disorder [29]. The General Self-Efficacy 10-item scale evaluates individual’s perceptions of their ability from not at all true (1) to exactly true (4), with summative scores ranging from 10 to 40 [30]. Memory was assessed through the memory satisfaction scale of the Multifactorial Memory questionnaire, an 18-item tool using a 5-point Likert scale scored 0–4 with some items reverse-scored [32,33]. The Multifactorial Memory questionnaire identifies total scores ranging from 0 to 72, with higher scores indicating greater memory [33]. Social health was assessed using the Social Support Index (SSI), including sub-scale scores for friend, family, and community support, perceptions of support, and overall social support [34]. This 17-item tool uses a 5-point Likert scale, with some questions reverse-scored, producing overall scores ranging from 0 to 68 and sub-scale scores of 0–12 (family, friend), 0–16 (community), and 0–20 (perceptions), with higher scores indicating stronger social support [34].

Physical fitness measures included cardiorespiratory fitness via the six-minute walk test, musculoskeletal strength through jump height, musculoskeletal power through leg power, flexibility through sit-and-reach flexibility, resistance strength via grip strength, and one-leg eyes-open and -closed balance tests according to standardized protocols [35,36]. The six-minute walk test included walking as far as possible across a 25 m hallway for 6 min [36]. Jump height was measured as the best of three jumps using a vertical jump trainer with participants touching the highest vane possible on each jump (Vertec^®^, sports Imports, Inc.; Columbus, OH, USA). Leg power was calculated from jump height and body mass (described below) [35]. Sit-and-reach flexibility of the hamstrings and lower back was measured with both feet flat against a wooden frame while participants reach, hand-over-hand, to push a wooden block along a ruler [35]. The best of three trials was recorded, with 26 cm or greater corresponding to reaching past participants’ feet [35]. Grip strength measured forearm muscular strength by squeezing a hydraulic hand dynamometer (Jamar, JLW Instruments; Chicago, IL, USA) and combining the greater of two left-hand trials with the greater of two right-hand trials [35]. Tandem stance eyes-open and -closed balance tests were also conducted according to standard procedures [37]. All balance tests were performed measuring the time participants could hold the stance without moving and were repeated twice on each leg or with each leg forwards and the best time of four trials was recorded [35,37]. All physical fitness tests indicate greater fitness with higher measurements.

Physical activity behavior was evaluated through the International Physical Activity Questionnaire [38]. This 27-question tool asks how many days per week and hours/minutes per day participants engage in moderate and vigorous physical activities as part of their work, as a form of transportation, around their house and as part of household responsibilities, and for their leisure time [38]. Days per week were multiplied by times reported for leisure time activity to calculate weekly moderate-to-vigorous physical activity. Physical health measures of waist circumference and body mass index (BMI) from body mass (kg) and height (cm) were also collected according to standard protocols [35]. Waist circumference was measured with a flexible tape measure at the iliac crest after inhalation [35]. Body mass was measured with shoes off, and height standing against a stadiometer (Health o meter^®^, Pelstar; McCook, IL, USA), with feet heels against the stadiometer and while inhaling [35]. Blood pressure was measured in a seated position after five minutes of seated rest and repeated three times, one minute apart (AD Cuff Adult 11, American Diagnostic Corporation; Hauppauge, NY, USA) [35]. The second and third measures of blood pressure were averaged to determine reported seated systolic and diastolic blood pressures. Measures of arterial stiffness including pulse-wave velocity (PWV), and neurovascular function of baroreceptor sensitivity (BRS), heart rate variability (HRV), and blood pressure variability (BPV) were collected using an electrocardiogram (ECG) along with simultaneous beat-by-beat pulse and blood pressure measurements (LabChart 8.1.22, ADinstruments Inc.; Colorado Springs, CO, USA) as previously utilized with Indigenous communities [39]. A three-lead ECG attached to electrode stickers on the right and left upper chest and right waist area measured heart rhythm. A finger photoplethysmography blood pressure cuff (Finapress Nova, Ohmeda, Inc.; Englewood, CO, USA) was placed between the distal and middle phalanges of the right middle finger to record beat-by-beat blood pressure and pulse waves at the brachial artery. Photoplethysmography sensors were placed on the carotid artery and the second right toe by the researcher. The participant placed the sensor for the femoral artery on the right medial femoral region with instruction from the researcher. These sensors measured pulse transit times with Powerlab 8/35 (ADinstruments Inc., Colorado Springs, CO, Colorado). Pulse-wave velocity was determined as measured distances of sensors over pulse transit times between the two sensors, including central PWV (carotid to femoral), upper peripheral PWV (carotid to finger), and lower peripheral PWV (femoral to toe). We evaluated HRV, BPV, and BRS via Nevrokard software (BRS–Baroreflex Sensitivity Analysis version 6.3.0, Nevrokard Inc.; Izola, Solvenia), including both sequence and spectral methods of BRS analysis.

### 2.4. Statistical Analysis

Comparisons of health measures before and after the intervention were evaluated through dependent *t*-test comparisons with *a priori* alpha set to 0.05. All assumptions of a dependent *t*-test were met, including assumptions of normality through analysis of skewness/kurtosis within normal limits, visual inspection of histograms, no outliers present, and randomly sampled participants [40]. Comparisons of health measures were evaluated overall, as well as by sex and Métis or non-Indigenous identities. To ensure anonymity of participants who reported genders other than that aligning with their sex at birth, analysis by gender was not conducted. Independent *t*-tests were used to compare demographic and health and wellbeing measures between participants who completed the study and those lost to follow-up.

## 3. Results

### 3.1. Pre-Intervention Findings

Of the 52 participants who attended pre-testing, 40 post-testing responses were collected. Age, sex, education, student status, and Red River Jigging experience were similar between participants who completed the program and post-intervention health assessments and those who were lost to follow-up. Measures of physical, cultural, mental, and social health, and physical fitness were similar between participants who completed the study and those lost to follow-up.

Table 1 details the participants’ age, identities, education, student status, and previous experience with the Red River Jig. Participants were similar across sexes and ethnicities, with primarily female participants and participants identifying as women. The majority of participants identified as non-Indigenous or Métis. Gender identity of participants was generally similar to their identified sex at birth, with two participants reporting gender identities not aligning with sex at birth. To ensure anonymity of participants, gender identity was not reported separate from sex. Of the 23 (57.5%) who returned tracking sheets for independent/at-home practice, 21 (91.3%) practiced outside of designated class time at least once per week, with 15 (65.2%) practicing twice per week, and 8 (34.7%) practicing more than twice per week.

### 3.2. Post-Intervention Findings

Figure 1 demonstrates the changes in the cultural and social health of participants after an 8-week Red River Jigging intervention. Ethnic identity (Figure 1A) remained unchanged among all participants, while ethnic affirmation and belonging (Figure 1B) declined among Métis (*p* = 0.04) participants. No changes in cultural exploration (Figure 1C) or cultural commitment (Figure 1D) were identified from pre-intervention to post-intervention overall, by sex or by ethnicity. Overall cultural identity (Figure 1E) also declined among Métis (*p* = 0.04) participants across the intervention. Increases in community support (Figure 1F) were also identified among female (*p* < 0.001), male (*p* = 0.02), Métis (*p* = 0.01), non-Indigenous (*p* = 0.004) and overall (*p* < 0.001) participants. Family support (Figure 1G) increased among female (*p* = 0.01) and overall (*p* = 0.046) participants. Friend support (Figure 1H) increased among female (*p* = 0.048) and overall (*p* = 0.04) participants. No differences in perceptions of social support (Figure 1I) or overall social support (Figure 1J) scores were identified.

From pre- to post-intervention, mental health measures of general self-efficacy (Figure 2A), wellbeing (Figure 2B), and mental wellbeing (Figure 2C) were not found to be different. Across the intervention, memory improved among male participants (Figure 2D; *p* = 0.048). Among physical health measures, systolic blood pressure (Figure 2E) declined among females (*p* = 0.02), Métis (*p* = 0.02), non-Indigenous (*p* = 0.047), and overall (*p* = 0.01) participants, though changes in diastolic blood pressure (Figure 2F) were not identified. Waist circumference and body mass index did not change overall or among any sub-groups with the intervention. Weight increased among male participants (Figure 2G; *p* = 0.04). No changes in central PWV (Figure 2H) and upper peripheral PWV (Figure 2I) were identified across the intervention. Lower peripheral PWV (Figure 2J) declined among non-Indigenous (*p* = 0.048) and overall (*p* = 0.04) participants.

Figure 3 outlines physical health measures of HRV, BPV, and BRS before and after the 8-week Red River Jig intervention overall and by sex and ethnicity. Mean HRV normal–normal intervals (Figure 3A) did not change with the intervention. Standard deviations of the HRV normal–normal intervals increased among Métis participants (Figure 3B, *p* = 0.02) after the intervention. Mean systolic BPV normal–normal intervals (Figure 3C) and standard deviations of normal–normal intervals (Figure 3D) did not change with the intervention. Diastolic BPV mean normal–normal intervals (Figure 3E) and standard deviations of normal–normal intervals (Figure 3F) also did not change with the intervention. Low-to-high-frequency HRV ratios did not change with the intervention (Figure 3G). Systolic blood pressure low-to-high-frequency ratios (Figure 3H) increased among male (*p* = 0.001) and Métis (*p* = 0.01) participants with the intervention. Spectral (Figure 3I) and sequence (Figure 3J) method BRS did not change from pre- to post-intervention.

Physical activity was not reported to change with the intervention (Figure 4A). Physical fitness improved across the intervention, including cardiorespiratory fitness (Figure 4B), among female (*p* = 0.002), Métis (*p* = 0.02), non-Indigenous (*p* = 0.04), and overall (*p* < 0.001) participants. From pre- to post-intervention, grip strength improved among male participants (Figure 4C; *p* = 0.02), while sit-and-reach flexibility improved among non-Indigenous participants (Figure 4D; *p* = 0.02). After the intervention, leg strength, measured by jump height (Figure 4E), improved among female (*p* = 0.002), Métis (*p* = 0.01), non-Indigenous (*p* = 0.04), and overall (*p* = 0.001) participants, while leg power (Figure 4F) increased among female (*p* = 0.04), Métis (*p* = 0.01), non-Indigenous (*p* = 0.04), and overall (*p* = 0.01) participants. One-leg eyes-closed balance improved among Métis participants (Figure 4G; *p* = 0.03), though one-leg eyes-open (Figure 4H), tandem eyes-closed (Figure 4I), and tandem eyes-open (Figure 4J) did not change across the intervention.

## 4. Discussion

### 4.1. Interpretation

The main purpose of this study was to investigate the effects of an 8-week Red River Jigging intervention on cultural, mental, social, and physical determinants of health. This study was the first to measure the health benefits of the Red River Jig, including benefits among both Métis and non-Indigenous Peoples. As a form of exercise, improvements in physical fitness and physical health were anticipated in this study [8]. Improvements in social health were significant benefits of this program, particularly the expanded support from the community established through this intervention. Further, the social health benefits experienced across Métis and non-Indigenous Peoples highlights the applicability of cultural activities beyond specific nations.

As the Red River Jig is a Métis dance, looking at the wholistic benefits of this culturally relevant activity may be important information for the community [4]. Declines in cultural affirmation and belonging and overall cultural identity scores, particularly among Métis participants, highlight the cultural nature of this dance. Across measures evaluated, ethnic identity, commitment, and exploration remained the same, while affirmation and belonging declined, leading to declines in overall cultural identity. Within the Red River Jig, steps and stories of the history, formation, and teachings of the Métis Nation are contained [10,41]. For Métis participants engaging in these classes, increased learnings and awareness of Métis history and culture may bring forth greater understanding of the cultural knowledge, teachings, and engagement participants do not currently hold, thus decreasing their feelings of affirmation and belonging within the culture, but maintaining their sense of cultural exploration, commitment, and identity. Further, many Métis People and families hid their Métis identity for generations to survive and may not have passed on cultural knowledge or practices to subsequent generations [42,43,44]. Subsequently, many Métis People have been reclaiming their identity and reengaging with their culture in recent decades [42,43,44]. A sense of not fitting in, which may be linked to cultural affirmation and belonging, has been common among Métis People in recent years [45]. The impacts of participation in this study on cultural affirmation and belonging of Métis People should be further explored. Changes in cultural identity scores among non-Indigenous participants were not expected as this intervention was grounded in Métis culture. In addition, there seemed to be sex-specific experiences that may be woven through cultural identities. Future studies should aim to include larger numbers of male participants to better evaluate sex-specific experiences of participation in such culturally grounded interventions.

Wellbeing, mental wellbeing, and self-efficacy did not change with the intervention. The lack of improvements in mental health may be due to the high health scores identified among these participants. Although there were no significant changes in mental health identified in this study, these are essential areas for future evaluation. Responses to the WHO-5 wellbeing questionnaire are similar to the 50–60th percentile among a large sample of people in Canada [46]. Similarly, mental wellbeing scores were above the 50th point, indicating moderately high wellbeing for this sample population [31]. The lack of significant changes in mental health in this study may be due to strong mental health among participants before the intervention, leaving little room for significant improvement to be detected. Similarly, improvements in memory satisfaction were only observed among male participants, where the greatest room for improvement was identified, and all other groups reported memory satisfaction consistent with average scores for similar aged populations [47].

Few studies have assessed vascular health among Indigenous populations [39]. Central PWV measured in the higher end of normal compared to similar age-referenced values [48]. Similarly, the standard deviation of the normal-to-normal (SDNN) measure for HRV among females in our study was in the 50th percentile of individuals, as reported in the previous literature [49]. Measures of BPV and BRS recorded in our participants were also similar to previous reference values [50]. Our female participants also were found to have lower systolic and diastolic blood pressures compared to adult reference populations [51], which may indicate our participants are more physically active, as supported by the reported physical activity among our participants [52]. Lower systolic low-to-high-frequency ratios, particularly below 1.33, are associated with greater likelihood of stroke and death [53]. Improved systolic low-to-high-frequency ratios, particularly among Métis participants in this study, where the lowest ratios were observed, indicates the intervention may be able to reduce risks of stroke and improve long-term outcomes for Métis People. As these measurements are highly sensitive to stress, medication, circadian variation, and other confounders, these results should be interpreted with caution; longer-term studies with regular follow-ups would be needed to accurately determine if stroke risks are reduced by participation in this study.

Social support among participants was very similar to reference populations, indicating that the sample population reflected the general population well in this aspect [54]. The improvement in self-perceived community support over this intervention could be due to the group component involved in the Red River Jig, possibly creating a community among participants. Especially for Indigenous populations, group activities are beneficial for wholistic health and may subsequently be more effective for improving health and wellbeing for Indigenous Peoples [55]. While no difference in feelings of community support between online and in-person communities was identified in the past, more recently, mental health has deteriorated with substitutions of in-person interactions with online interactions [56,57]. As the Red River Jig intervention in this study offered a hybrid model including in-person or virtual options to participate, improvement in feelings of community support and no changes in mental health measures are important findings.

Decreases in systolic blood pressure and increases in leg power, even with pre-intervention blood pressures in the healthy range and leg power in the top 20th percentile, indicate health benefits from Red River Jig even among fit, healthy individuals [51,58]. Improvements in health measures across social, mental, and physical health, and physical fitness highlight the wholistic health and wellbeing benefits of this intervention. Further, the improvements in social and physical health and physical fitness among both sexes as well as both Métis and non-Indigenous participants highlight the applicability of this intervention for all adults.

### 4.2. Strengths

The strengths of this study include evaluating the wholistic health benefits of a cultural form of physical activity, using validated measurement tools and clinically validated tests. Multiple interventions were conducted to ensure a large enough sample size and many opportunities for individuals were provided to participate in this exciting form of physical activity, and contributed to long-term sustainability and learning as participants had opportunities to continue attending classes after their participation in the research study had concluded. This study also provided opportunities for members of the Métis community in Saskatoon to engage in Red River Jigging beyond the scope of this study. Future studies in this area should engage large sample sizes and expand to include older individuals who may experience poorer health and wellbeing prior to the intervention, to further evaluate wholistic health benefits of this dance.

### 4.3. Limitations

Limitations of this study include the pre-experimental design, lacking a control group and the heterogeneous, non-random sample engaged in the intervention. The inclusion of a control group might have provided better understanding and quantitative measurement of the impacts of this study. However, as directed by the community partner, a control group was not supported, as the community felt it was valuable to provide dance opportunities for as many participants as possible, and preservation and transmission of culture was held as a higher priority. The small number of males engaged in this study is consistent with other adult group physical activity programs among Indigenous communities [59]. Due to this low sample of males, the effectiveness of the program to improve health and wellbeing among males is limited. Some participants’ at-home practice was recorded, but many participants did not record or return records of their independent practice, and other forms of physical activity during the intervention were not measured. The high mental wellbeing of participants in this study may have limited the ability to measure benefits of mental wellbeing resulting from this study. Similarly, participants were found to have generally healthy measures of physical function, physical fitness, and social support, which may have limited the improvements identified through this program.

### 4.4. Future Research

Future research programs should evaluate similar culturally grounded physical activity programs among diverse participants, including those at greater risk of chronic disease and less strong mental wellbeing. Exploring similar Métis dancing programs among males and among older adults would be valuable. Future studies could also consider classes offered at multiple times per week to engage participants in group training three times per week and reduce the attrition rate by meeting more participants’ availability needs. With community partnership, future studies might consider engaging a multiple-armed intervention trial to compare Red River Jigging, or other Métis dances, with another cultural activity such as beading or sash weaving. Future studies should also engage qualitative evaluation of cultural connection and experience of Métis participants in interventions such as these to better explore why cultural identity declined with participation in a Métis culturally grounded program.

## 5. Conclusions

A Red River Jig intervention provides a safe and effective form of exercise that can benefit multiple dimensions of health among male, female, Métis and non-Indigenous populations, including physical health, physical fitness, and social health. Future research, including a larger sample size or more Métis People, should be evaluated to further examine improvements in cultural and psychosocial dimensions with a Red River Jig intervention.

## Figures and Tables

**Figure 1 ijerph-22-01225-f001:**
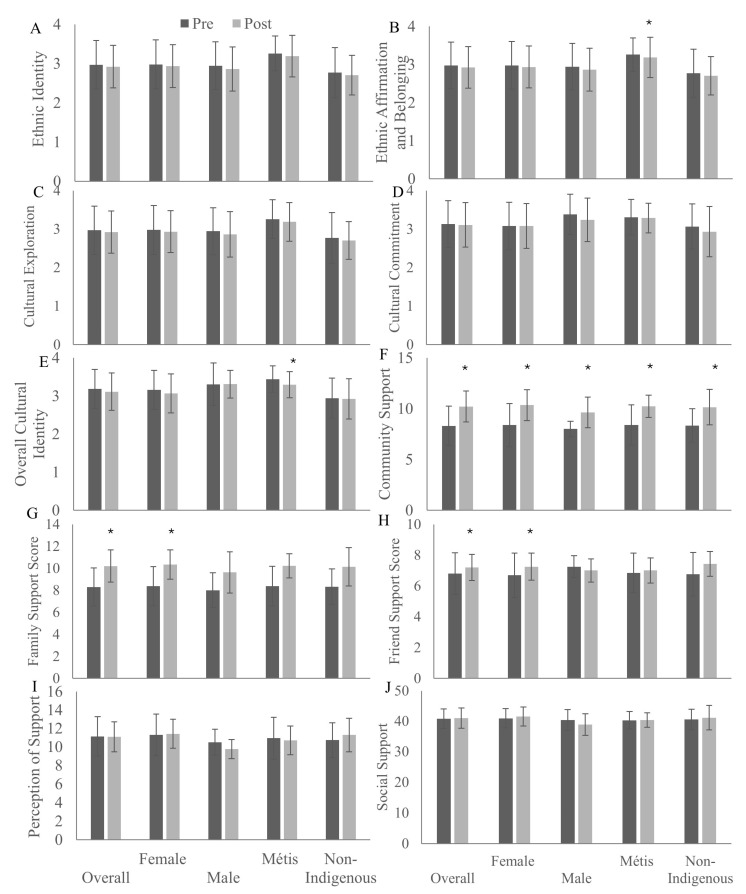
Cultural connectedness: ethnic identity (**A**), ethnic affirmation (**B**), ethnic exploration (**C**), ethnic commitment (**D**), cultural identity (**E**), and social support: community (**F**), family (**G**), friend (**H**), perceptions of (**I**), and overall (**J**) scores of participants before and after an 8-week Red River Jigging intervention in Saskatchewan, 2022–2025. * indicates significant change from pre-intervention, *p* < 0.05.

**Figure 2 ijerph-22-01225-f002:**
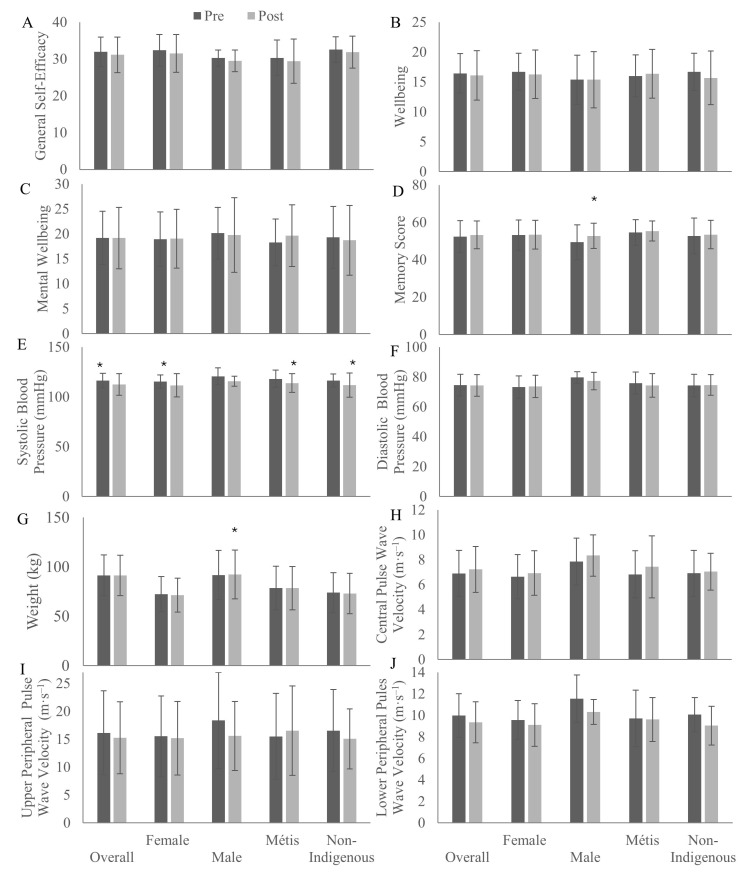
Mental wellbeing scores of general self-efficacy (**A**), wellbeing (**B**), mental wellbeing (**C**), and memory (**D**), and physical health measures of resting systolic blood pressure (**E**), resting diastolic blood pressure (**F**), weight (**G**), central pulse wave velocity (**H**), upper peripheral pulse wave velocity (**I**), and lower peripheral pulse wave velocity (**J**) of participants before and after an 8-week Red River Jigging intervention in Saskatchewan, 2022–2025. * indicates significant change from pre-intervention, *p* < 0.05.

**Figure 3 ijerph-22-01225-f003:**
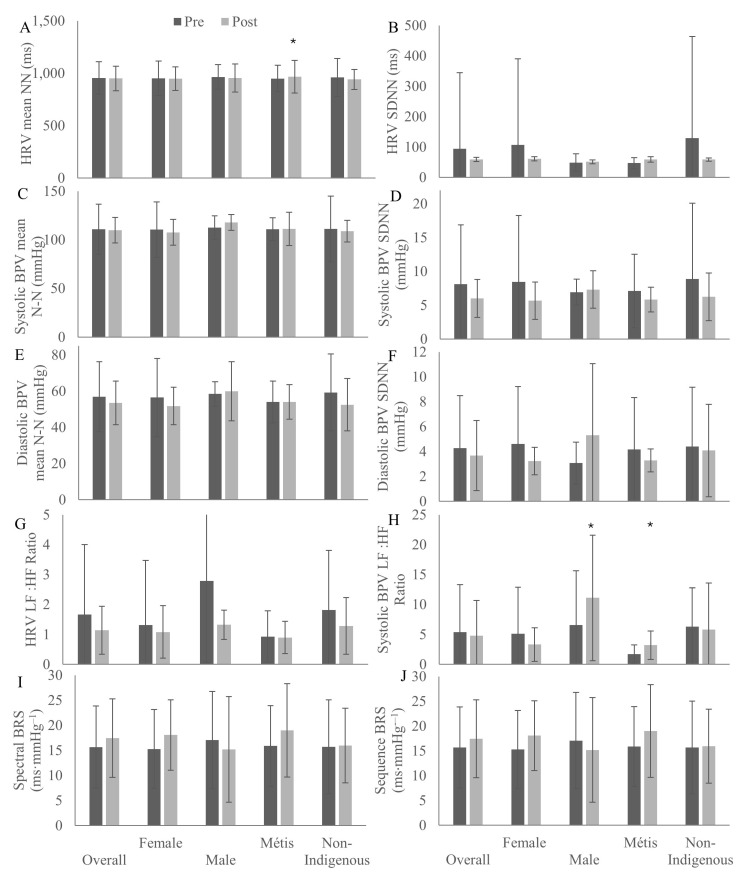
Heart rate variability (HRV), blood pressure variability (BPV) and baroreceptor sensitivity (BRS) measures of HRV mean beat-to-beat interval (NN) (**A**), HRV standard deviation of mean NN (SDNN) (**B**), systolic BPV mean NN (**C**), systolic BPV SDNN (**D**), diastolic BPV mean NN (**E**), diastolic BPV SDNN (**F**), HRV low frequency (LF):high frequency (HF) ratio (**G**), systolic BPV LF:HF ratio (**H**), spectral BRS (**I**), and sequence BRS (**J**) of participants before and after an 8-week Red River Jigging intervention in Saskatchewan, 2022–2025. * indicates significant change from pre-intervention, *p* < 0.05.

**Figure 4 ijerph-22-01225-f004:**
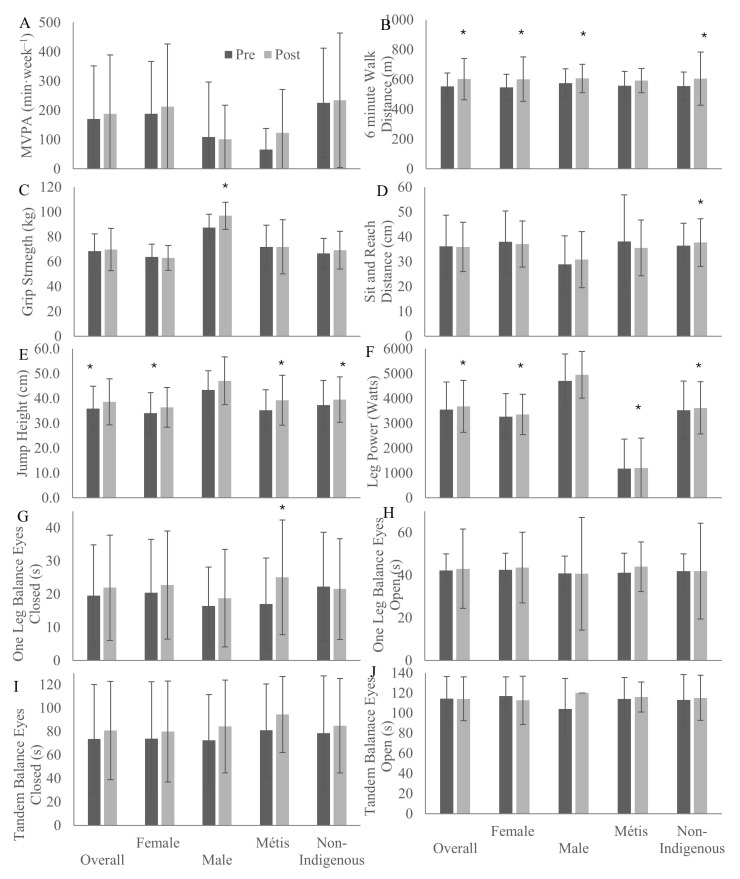
Physical activity and physical fitness measures of weekly leisure moderate-to-vigorous physical activity (MVPA; **A**), six minute walk distance (**B**), grip strength (**C**), sit and reach flexibility (**D**), jump height (**E**), leg power (**F**), one leg eyes closed balance (**G**), one leg eyes open balance (**H**), tandem eyes closed balance (**I**), and tandem eyes open balance (**J**) of participants before and after an 8-week Red River Jigging intervention in Saskatchewan, 2022–2025. * indicates significant change from pre-intervention, *p* < 0.05.

**Table 1 ijerph-22-01225-t001:** Demographic characteristics of Red River Jig program participants in Saskatchewan 2022–2025, by participant ethnicity and sex, n (%), mean ± SD.

	Female	Male	Métis	Non-Indigenous	Overall
Completed Study, n (%)	32 (76.2)	8 (80.0)	14 (77.8)	20 (71.4)	40 (76.9)
Age, mean ± SD	39 ± 16	40 ± 13	39 ± 16	39 ± 15	39 ± 15
Male, n (%)	0 (0.0)	8 (100.0)	3 (21.4)	5 (25.0)	8 (20.0)
Woman †, n (%)	30 (93.8)	0 (0.0)	11 (78.6)	13 (65.0)	30 (75.0)
Man †, n (%)	0 (0.0)	8 (100.0)	3 (21.4)	5 (25.0)	8 (20.0)
Métis, n (%)	11 (34.4)	3 (37.5)	14 (100.0)	0 (0.0)	14 (35.0)
First Nations, n (%)	3 (9.4)	0 (0.0)	0 (0.0)	0 (0.0)	3 (7.5)
Non-Indigenous, n (%)	15 (46.9)	5 (62.5)	0 (0.0)	20 (100.0)	20 (50.0)
Education					
High School or Less, n (%)	6 (18.8)	0 (0.0)	4 (28.6)	1 (5.0)	6 (15.0)
Trade School or College, n (%)	11 (34.4)	3 (37.5)	4 (28.6)	5 (25.0)	14 (35.0)
At Least Some University, n (%)	15 (46.9)	5 (62.5)	6 (42.9)	14 (70.0)	20 (50.0)
Full-time Student, n (%)	15 (46.9)	3 (37.5)	5 (35.7)	11 (55.0)	18 (45.0)
Prior Experience with Red River Jig, n (%)	7 (25.0)	1 (10.0)	5 (35.7)	2 (10.0)	8 (20.0)
Number of Classes Attended, mean ± SD	7.7 ± 1.1	8.0 ± 0.5	7.7 ± 1.0	7.9 ± 0.8	7.8 ± 1.0

† Other genders not reported due to small sample sizes; n, number of participants; SD, standard deviation.

## Data Availability

The participants of this study, and the community partners, did not give written consent for their data to be shared publicly. Aligning with Métis Data Governance Principles, this data are owned and access to these data is determined by the partnering Métis community. As such, the researchers do not have authority to share this data, and the community partner cannot commit to having future capacity to evaluate and respond to data access requests; as such, supporting data are not available.

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
