# Peer review of "The Effects of the Red River Jig on the Wholistic Health of Adults in Saskatchewan"

_ijerph, 2025, doi:10.3390/ijerph22081225_

Round 1
Reviewer 1 Report
Comments and Suggestions for Authors
Thank you for allowing me to review the manuscript entitled: The effects of the Red River Jig on the wholistic health of adults in Saskatchewan.
The article is well written. My major comments relate to the explanation of the methods and ethical approval. Please refer to my notes below.
- What needs to be better explained for the audience:
- Lines 56-58: Your readers might not be familiar with the dance. They will be from all over the world. Can you better describe it than it is a fiddle dance? Is there twirling? Do men and women dance separately? How can you convey the essence of the dance in words for those who are unfamiliar with it?
- Lines 58-60: Jigging allows all community members, including Europeans and non-Indigenous Peoples, to participate in gatherings, bridging community divides (Quick, 2008).
I believe this describes how the dance originated, and then it was European, such as the French and Scots. You will need to change the sentence to the past tense. I suggest you describe the importance of the dance at the time it originated.
- In several places, you refer to ‘community members’. What is community for this study? Is it the Metis community that participated in the study, or is it the community of people (all of them) that took the dance lessons? I suggest you clarify this.
- The knowledge gap is not justified by literature or by Metis knowledge.
Lines 60-63: Métis communities have long recognized the wholistic benefits of jigging related to the physical activity along with connection to culture and community; however, westernized knowledge and understanding of the impact of jigging has not yet been shared/published in a colonized form.
Is there literature to support this claim? If yes, it should be cited.
If not, there might be Elders knowledge to support this, perhaps in the TRC.
In the way this is presented, it is your opinion, and this is not enough for a research paper.
- There is no description of context
In section 1, before you describe the methods, there should be a description of the context. What is Saskatchewan? Where is it? Why there? Are there many Metis peoples there ir was it just convenient?
- Methods
- Lines 69-73: The study adopted the Métis infinity symbol worldview to seek the effectiveness of the Red River Jig as a physical activity intervention (LaVallee, 2014). Constant collaboration and communication with partnering Li Toneur Niimiyitoohk Métis dance group ensured the partnering Métis communities’ values were prioritized throughout the study. Community Advisors from Li Toneur Niimiyitoohk Métis dance group were involved throughout the research process
You need to explain your framework better. Why did you adopt this specific framework? What is the framework? Why are collaboration and communication with your research partners important? What is the role of community advisors? Why do you have them?
- While there is a sentence that says the study received ethics approval, there is no evidence that this study received ethics’ approval. REBs provide letters with a number indicating their approval, this is missing and it should be added. The Metis Nation has its own principles of ethical research. You need to describe how this research adheres to them, mainly because TCPS 2 (2022), Chapter 9, is very precise about how to conduct research ethically with Métis peoples. How did you engage the group to conduct research with them?
- How were participants recruited and who recruited them? Were the participants students of the University or were non-students included as well as participants? Did they receive payment to participate? Why yes and why not? Participants had very invasive procedures done on them, such as an ECG. There is no description about how these procedures were done; was it before the dance classes? Refer to the guidelines for research with the Métis Nations regarding payment.
- You obtained data on gender identity. Was a gender analysis of the data performed? I do not see indications about this, in lines 143 – 144 you mention analysis done only by sex. It would be interesting to explore gender for jigging from a cultural and social health perspective.
- Discussion
Lines 220- 222: you refer to ethnic identity remaining the same, while affirmation and belonging declined, leading to declines in overall cultural identity. This is nicely discussed in the sentences that follow. This is a significant finding, and I do not see it brought forward in the abstract.
Overall, there have to be significant improvements in the methods section, as outlined, and a better explanation of the ethics approval. The other suggested changes would make the manuscript better for a global readership.
Author Response
Dear Reviewer:
We would like to thank you sincerely for your considerate review of our submission ‘“Community traditions, community kinship, language, and land bring me a lot of joy”: The Importance of Culture and Social Support in the Health of Métis People’. As outlined below, to the best of our ability, all of your suggested edits have been made in the revised version of this manuscript. Revisions to the manuscript have also been completed to address each of your comments and feedback. Thank you again for your consideration, your comments and feedback have been greatly appreciated. We feel that these recommended changes have strengthened our manuscript greatly.
Thank you for allowing me to review the manuscript entitled: The effects of the Red River Jig on the wholistic health of adults in Saskatchewan.
The article is well written. My major comments relate to the explanation of the methods and ethical approval. Please refer to my notes below.
- What needs to be better explained for the audience:
Comment 1: Lines 56-58: Your readers might not be familiar with the dance. They will be from all over the world. Can you better describe it than it is a fiddle dance? Is there twirling? Do men and women dance separately? How can you convey the essence of the dance in words for those who are unfamiliar with it?
Response 1: Thank you for this comment. We have added further information describing the dance.
Comment 2: Lines 58-60: Jigging allows all community members, including Europeans and non-Indigenous Peoples, to participate in gatherings, bridging community divides (Quick, 2008).
I believe this describes how the dance originated, and then it was European, such as the French and Scots. You will need to change the sentence to the past tense. I suggest you describe the importance of the dance at the time it originated.
Response 2: Thank you for this suggestion. We have added further details about the origin and importance of this dance, and updated to reflect past tense.
Comment 3: In several places, you refer to ‘community members’. What is community for this study? Is it the Metis community that participated in the study, or is it the community of people (all of them) that took the dance lessons? I suggest you clarify this.
Response 3: Thank you for pointing this out, we have revised the wording throughout to clearly indicate who is being referred to where we previously said “community members”
Comment 4: The knowledge gap is not justified by literature or by Metis knowledge. Lines 60-63: Métis communities have long recognized the wholistic benefits of jigging related to the physical activity along with connection to culture and community; however, westernized knowledge and understanding of the impact of jigging has not yet been shared/published in a colonized form.
Is there literature to support this claim? If yes, it should be cited.
If not, there might be Elders knowledge to support this, perhaps in the TRC.
In the way this is presented, it is your opinion, and this is not enough for a research paper.
Response 4: Thank you for this suggestion. We have added further details about the historical and contemporary knowledge about wholistic benefits of jigging.
Comment 5: There is no description of context. In section 1, before you describe the methods, there should be a description of the context. What is Saskatchewan? Where is it? Why there? Are there many Metis peoples there ir was it just convenient?
Response 5: Thank you for this suggestion. We have added details to the start of the methods to describe Saskatchewan and the Métis population.
Comment 6: Lines 69-73: The study adopted the Métis infinity symbol worldview to seek the effectiveness of the Red River Jig as a physical activity intervention (LaVallee, 2014). Constant collaboration and communication with partnering Li Toneur Niimiyitoohk Métis dance group ensured the partnering Métis communities’ values were prioritized throughout the study. Community Advisors from Li Toneur Niimiyitoohk Métis dance group were involved throughout the research process
You need to explain your framework better. Why did you adopt this specific framework? What is the framework? Why are collaboration and communication with your research partners important? What is the role of community advisors? Why do you have them?
Response 6: Thank you for this opportunity to expand on the Indigenous research methods and approach engaged in this project. We have added further details around the meaning of this framework.
Comment 7: While there is a sentence that says the study received ethics approval, there is no evidence that this study received ethics’ approval. REBs provide letters with a number indicating their approval, this is missing and it should be added. The Metis Nation has its own principles of ethical research. You need to describe how this research adheres to them, mainly because TCPS 2 (2022), Chapter 9, is very precise about how to conduct research ethically with Métis peoples. How did you engage the group to conduct research with them?
Response 7: Thank you for this opportunity to expand on the Indigenous research methods and approach engaged in this project. We have added further details around the engagement of Métis research ethics and TCPS 2 Chapter 9.
Comment 8: How were participants recruited and who recruited them? Were the participants students of the University or were non-students included as well as participants? Did they receive payment to participate? Why yes and why not? Participants had very invasive procedures done on them, such as an ECG. There is no description about how these procedures were done; was it before the dance classes? Refer to the guidelines for research with the Métis Nations regarding payment.
Response 8: Thank you for identifying this missing information. We have added information about participant inclusion and exclusion criteria, honoraria provided, and location of these health assessments.
Comment 9: You obtained data on gender identity. Was a gender analysis of the data performed? I do not see indications about this, in lines 143 – 144 you mention analysis done only by sex. It would be interesting to explore gender for jigging from a cultural and social health perspective.
Response 9: Thank you for this comment. We agree gender analysis of data would be an interesting perspective. As two participants reported gender identity not aligning with sex at birth, and one or more of these identities as gender diverse (i.e., not man or woman), we felt it would be a risk to anonymity of these participants to report their gender identity in the manuscript. With a larger sample size in future work, we see value in analyzing data by gender identity in future.
Comment 10: Lines 220- 222: you refer to ethnic identity remaining the same, while affirmation and belonging declined, leading to declines in overall cultural identity. This is nicely discussed in the sentences that follow. This is a significant finding, and I do not see it brought forward in the abstract.
Response 10: Thank you for this suggestion. We have added a sentence highlighting this finding to the abstract: Ethnic identity remained the same while affirmation and belonging declined, leading to declines in overall cultural identity, as learning about Métis culture through the Red River jig may highlight gaps in cultural knowledge.
Comment 11: Overall, there have to be significant improvements in the methods section, as outlined, and a better explanation of the ethics approval. The other suggested changes would make the manuscript better for a global readership.
Response 11: Thank you for this summary. We have made many additions to the methods to better explain the context, ethical process engaged, and specific details of testing sessions and honoraria. We are happy to make further changes if the reviewers feel it appropriate.
Reviewer 2 Report
Comments and Suggestions for Authors
Dear authors, I'm leaving you a Word document with my comments, questions, and suggestions.
All with love.

Author Response
Dear Reviewer:
We would like to thank you sincerely for your considerate review of our submission ‘“Community traditions, community kinship, language, and land bring me a lot of joy”: The Importance of Culture and Social Support in the Health of Métis People’. As outlined below, to the best of our ability, all of your suggested edits have been made in the revised version of this manuscript. Revisions to the manuscript have also been completed to address each of your comments and feedback. Thank you again for your consideration, your comments and feedback have been greatly appreciated. We feel that these recommended changes have strengthened our manuscript greatly.
Dear researchers:
I liked your work. I greatly appreciate finding different ways to promote physical activity, especially when it has such a strong cultural component. I hope my comments are taken positively.
By improving the materials and methods, the limitations, and perhaps updating the references, it could be a 10 out of 10.
To be honest, I don’t know much about the topic, but from a scientific perspective, I’m concerned about the predominance of citations more than 15 years old in your work. We need to find ways to improve this. While there’s likely little literature available due to the very specific topic, a solution needs to be found.
Comment 1: When you start the “Spiritual Aspects” paragraph, you should mention that the definitions are based on Metis culture. If they were general, you could use more current references to support the idea.
Response 1: Thank you for this suggestion. We have added a few sentences to frame this section with reference to Métis culture and approaches to health.
Comment 2: The statement “Regular physical activity has many health benefits, such as weight and blood pressure reductions, and feelings of mental relief and positivity (Warburton et al., 2006) must be accompanied by an updated reference, and there are too many. The paragraph is very good overall; the only difficulty is what I mentioned about the recentness of the quotes. The objective and hypothesis to conclude the section are perfect.
Response 2: Thank you for this suggestion, an updated reference was added (Mahindru et al., 2023).
Comment 3: I think it’s good to add that the participation was voluntary.
Response 3: Thank you for this suggestion, we have added voluntary to describe the consent participation.
Comment 4: A section on inclusion and exclusion criteria is missing. It doesn’t specific whether the person is over 18, male, or female. A little more information needs to be added so the study can be replicated in the future.
Response 4: Thank you for highlighting this omission. We have added details of inclusion and exclusion criteria: Participants were eligible to participate if they were at least 18 years old, did not have a history of heart disease (heart attack, stroke, bypass surgery, angina, etc.) or diabetes, and were cleared for unrestricted physical activity. Participants did not need to be a member of Li Toneur Niimiyitoohk Métis Dance Group to participate in this study.
Comment 5: The same applies to the exclusion criteria: some pathology or exercise restriction, I don’t know, but some detail should be included.
Response 5: See above response to Comment 4.
Comment 6: Why, if it’s 8 weeks, is there such a large timeframe (2022-2025)? Please answer or elaborate.
Response 6: We have added additional information. A total of 8 interventions were held to accomplish the a priori target sample size of 40 participants planned to enable sub-group analysis of blood pressures.
Comment 7: Were the independent home practices the same length? Were you able to confirm this in any way? Remember that in PA, the timing and frequency of interventions are very important. I have a question: Did the participants complete an in-person session and practice independently for two days, achieving a frequency of three days a week?
Response 7: Thank you for identifying this omission. We have added data about independent practice sessions from the participants who returned logs of their practice sessions.
Comment 8: In some assessments, you clearly mentioned the measurement method or the object you used to measure, while in others you didn’t. You need to unify the approach. Also, talk a little about data interpretation. What does each test tell you? Is it scored on a scale from 1 to 5, from 10 to 20? Is a higher or lower score better? Do you get the idea?
Response 8: We have added additional details for each measurement including details of the questionnaire and make/model of equipment.
Comment 9: Were the tests taken at the university? In a lab? On separate days? All together? There are no details of how they will present their results.
Response 9: Thank you for identifying this omission, we have added details of participant tests, on separate days at the University of Saskatchewan campus in Merlis Belsher Place.
Comment 10: Especially in physical activity tests, it is necessary to detail exactly what was used to measure, since depending on what it is, there may be a margin of error or different values.
Response 10: We have added the details of equipment used for each test, including make and model, and manufacturer.
Comment 11: The results and explanations are consistent, and the analysis is accurate.
I don’t have much to add; visually, I’m not 100% convinced that there are so many figures, but they measured several variables, so all in all, it’s fine.
Response 11: Thank you for this comment, it is quite a few figures, but as you identified, quite a few measures.
Comment 12: I think it’s more accurate to explicitly state the same objective you stated in the introduction to begin the discussion. This section is by far the most solid part of their research; they addressed and justified the reasons for all their results. Very well done
Response 12: We have adjusted this sentence to include the same objective stated in the introduction replaced the modified objective in the beginning of the discussion.
Comment 13: I think it is important to separate limitations, strengths, practical implications and future lines of research.
There is no limitations section, and there are several: their study is pre-experimental, meaning there is no control group; they have a sample with a smaller proportion of men and are not random. They do not know the duration of the home sessions or whether they were actually performed. They do not know if the results are the same for men and women. There is no control over fatigue or physical levels outside of practice sessions. A high level of mental health can be a limitation that prevents observing significant changes.
Response 13: Thank you for identifying this gap. We have added a limitations section, including the limitations identified.
Comment 14: There is no recommendation on protocol or how to learn this dance.
Response 14: We have added a section on future recommendations for similar or subsequent research studies. We have not added recommendations on the most effective ways of learning this dance as participants’ proficiency in the dance was beyond the scope of this project.
Comment 15: Of the 52 participants, they collected 40 post-tests, which means they still lost a significant amount of information.
Response 15: We recognize the high attrition rate of participants. Due to limited options for the intervention classes (1-2 times per week), some participants who attended pre-intervention health assessment testing were not available to attend the jigging classes and were thus unable to continue in the study. We also experienced some participant withdrawals after participants attended 1-2 jigging classes and a few participants who attended classes but chose not to attend or were unavailable for post-intervention health assessment testing.
Comment 16: The conclusion is perfect, concrete, concise and clear based on its results.
Response 16: Thank you for this comment
Reviewer 3 Report
Comments and Suggestions for Authors
This manuscript presents a unique and meaningful contribution to the literature by investigating the health benefits of the Red River Jig, a traditional Indigenous dance. Its originality lies in the way it connects culturally grounded practices with established biomedical indicators of health, an intersection that remains significantly underexplored in current research.
The authors also demonstrate a strong and respectful engagement with the Métis community. Their efforts to prioritize cultural partnership, involve community advisors, and co-develop the intervention reflect an ethically sound and socially responsible approach to research.
Another notable strength is the comprehensive scope of the health outcomes examined. The inclusion of psychosocial, physical, cultural, and vascular dimensions of health enriches the study’s relevance and aligns well with wholistic conceptions of well-being often emphasized in Indigenous worldviews.
Furthermore, the use of validated and standardized assessment tools lends credibility to the findings. Measurements related to physiological function, physical fitness, and mental well-being are all grounded in established protocols, which strengthens the methodological rigor of the study.
Finally, the presentation of the results—particularly through the use of well-designed figures and tables—enhances readability and facilitates interpretation. The visual clarity of the data supports the overall coherence of the manuscript.
In another hand,
there are several key limitations that should be addressed to strengthen the scientific validity and impact of this work.
First and foremost, the absence of a control group is a major limitation. Without a comparison group, it becomes difficult to isolate the effects of the intervention from other potential influences, such as time, seasonality, or participant expectations.
The small sample size (N=40) and its strong female predominance (80%) also pose challenges to generalizability. While the authors acknowledge this demographic imbalance, no statistical adjustments or subgroup sensitivity analyses are offered to mitigate its impact.
A particularly intriguing finding—the decline in cultural identity scores among Métis participants—is noted but not thoroughly explored. This counterintuitive result deserves deeper theoretical reflection or at least some qualitative inquiry, as it raises important questions about cultural self-perception in the context of re-engagement with traditional practices.
Another concern is the loss of participants: 12 out of 52 individuals dropped out before post-testing. Unfortunately, the manuscript does not include a sensitivity analysis to assess whether these dropouts might have influenced the outcomes.
Additionally, some physiological results—such as those related to HRV and BPV—are interpreted with confidence despite the relatively modest sample size. Given the complexity of these biomarkers and their susceptibility to external confounders (e.g., stress levels, medication use, circadian variation), more caution in interpretation would be appropriate.
Lastly, while ethical approval and informed consent are clearly stated, the explanation for why participant data cannot be shared (“due to sensitivity”) is vague. In the context of open science, the authors should consider offering anonymized datasets or providing a more specific justification for withholding data.
Comments on the Quality of English LanguageRegarding language and structure, while the manuscript is largely readable, several sentences could benefit from improved clarity and conciseness. Minor editorial revision is recommended to enhance flow and avoid ambiguity.
Author Response
Dear Reviewer:
We would like to thank you sincerely for your considerate review of our submission ‘“Community traditions, community kinship, language, and land bring me a lot of joy”: The Importance of Culture and Social Support in the Health of Métis People’. As outlined below, to the best of our ability, all of your suggested edits have been made in the revised version of this manuscript. Revisions to the manuscript have also been completed to address each of your comments and feedback. Thank you again for your consideration, your comments and feedback have been greatly appreciated. We feel that these recommended changes have strengthened our manuscript greatly.
This manuscript presents a unique and meaningful contribution to the literature by investigating the health benefits of the Red River Jig, a traditional Indigenous dance. Its originality lies in the way it connects culturally grounded practices with established biomedical indicators of health, an intersection that remains significantly underexplored in current research.
The authors also demonstrate a strong and respectful engagement with the Métis community. Their efforts to prioritize cultural partnership, involve community advisors, and co-develop the intervention reflect an ethically sound and socially responsible approach to research.
Another notable strength is the comprehensive scope of the health outcomes examined. The inclusion of psychosocial, physical, cultural, and vascular dimensions of health enriches the study’s relevance and aligns well with wholistic conceptions of well-being often emphasized in Indigenous worldviews.
Furthermore, the use of validated and standardized assessment tools lends credibility to the findings. Measurements related to physiological function, physical fitness, and mental well-being are all grounded in established protocols, which strengthens the methodological rigor of the study.
Finally, the presentation of the results—particularly through the use of well-designed figures and tables—enhances readability and facilitates interpretation. The visual clarity of the data supports the overall coherence of the manuscript.
In another hand,
there are several key limitations that should be addressed to strengthen the scientific validity and impact of this work.
Comment 1: First and foremost, the absence of a control group is a major limitation. Without a comparison group, it becomes difficult to isolate the effects of the intervention from other potential influences, such as time, seasonality, or participant expectations.
Response 1: We recognize the limitation of not engaging a control group in a physical activity intervention. We have added a limitation section and expanded on this limitation as well as provided potential alternatives to a control group in a future considerations section. Due to the community direction and the priority in maintaining and transmitting culture, our community partner was not in support of a control group for this study. These details have been described in the limitations section.
Comment 2: The small sample size (N=40) and its strong female predominance (80%) also pose challenges to generalizability. While the authors acknowledge this demographic imbalance, no statistical adjustments or subgroup sensitivity analyses are offered to mitigate its impact.
Response 2: We ran analysis to compare the participants who completed the intervention and those who were lost to follow-up. None of the measures of the study or demographics were different between these participants.
Comment 3: A particularly intriguing finding—the decline in cultural identity scores among Métis participants—is noted but not thoroughly explored. This counterintuitive result deserves deeper theoretical reflection or at least some qualitative inquiry, as it raises important questions about cultural self-perception in the context of re-engagement with traditional practices.
Response 3: We have expanded the discussion to highlight the history of hidden Métis identity and how this may contribute to reduced feelings of belonging and affirmation for Métis People. We have also highlighted the need to explore this experience in qualitative studies in the future directions section.
Comment 4: Another concern is the loss of participants: 12 out of 52 individuals dropped out before post-testing. Unfortunately, the manuscript does not include a sensitivity analysis to assess whether these dropouts might have influenced the outcomes.
Response 4: We recognize the high attrition rate of participants. Due to limited options for the intervention classes (1-2 times per week), some participants who attended pre-intervention health assessment testing were not available to attend the jigging classes and were thus unable to continue in the study. We also experienced some participant withdrawals after participants attended 1-2 jigging classes and a few participants who attended classes but chose not to attend or were unavailable for post-intervention health assessment testing. We ran analysis to compare the participants who completed the intervention and those who were lost to follow-up. None of the measures of the study or demographics were different between these participants.
Comment 5: Additionally, some physiological results—such as those related to HRV and BPV—are interpreted with confidence despite the relatively modest sample size. Given the complexity of these biomarkers and their susceptibility to external confounders (e.g., stress levels, medication use, circadian variation), more caution in interpretation would be appropriate.
Response 5: Thank you for this caution, we have added this caution to the interpretation and added the need for longer term follow up to accurately determine if stroke risks are reduced by participation in this study.
Comment 6: Lastly, while ethical approval and informed consent are clearly stated, the explanation for why participant data cannot be shared (“due to sensitivity”) is vague. In the context of open science, the authors should consider offering anonymized datasets or providing a more specific justification for withholding data.
Response 6: Thank you for this comment. We have added additional details to describe the data ownership and authority for access to data held by the Indigenous community partner. The community partner cannot commit to capacity to evaluating and responding to future requests for access to their data and subsequently are not able to offer access to the data. While we appreciate this is not in line with open science approaches becoming more common, it is consistent with Indigenous data ownership and control, which does not rest with the researchers.
Comment 7: Comments on the Quality of English Language. Regarding language and structure, while the manuscript is largely readable, several sentences could benefit from improved clarity and conciseness. Minor editorial revision is recommended to enhance flow and avoid ambiguity.
Response 7: A few minor revisions were made to grammar and sentence structure.
Round 2
Reviewer 1 Report
Comments and Suggestions for Authors
The manuscript has improved significantly since the first submission, and you have explained the concepts nicely. I only suggest adding that gender analysis was not possible to be done given your explanations in Comment 9.
Well done.
Author Response
Thank you for your subsequent review of our manuscript. We appreciate your feedback and suggestions to improve the manuscript.
Comment 1:
The manuscript has improved significantly since the first submission, and you have explained the concepts nicely. I only suggest adding that gender analysis was not possible to be done given your explanations in Comment 9.
Well done.
Response 1: We have added further details regarding the absence of gender-based analysis to the statistical analysis section of the methods.
Reviewer 2 Report
Comments and Suggestions for Authors
The document has improved substantially, particularly regarding the methodology and the updates to the references in the introduction.
I think my final point for fully accepting your manuscript is to include subheadings for the methodology to make it easier to read, for example: study design, intervention or protocol, participants, inclusion criteria, exclusion criteria, data collection, just as you included the data analysis. Order it however you like.
The same applies to the final section: subheadings for limitations and future studies. You could even add one for practical implications.
Author Response
Thank you for your subsequent review of our manuscript. We appreciate your feedback and suggestions to improve this manuscript.
Comment 1:
The document has improved substantially, particularly regarding the methodology and the updates to the references in the introduction.
I think my final point for fully accepting your manuscript is to include subheadings for the methodology to make it easier to read, for example: study design, intervention or protocol, participants, inclusion criteria, exclusion criteria, data collection, just as you included the data analysis. Order it however you like.
The same applies to the final section: subheadings for limitations and future studies. You could even add one for practical implications.
Response 1: Thank you for this suggestion. We have added appropriate headings for the methods and discussion sections.
Reviewer 3 Report
Comments and Suggestions for Authors
In my opinion, the manuscript fit all criteria to be published by IJERPH.
Author Response
In my opinion, the manuscript fit all criteria to be published by IJERPH.
Thanks.